# Right Ventricular Strain and Left Ventricular Strain Using Speckle Tracking Echocardiography—Independent Prognostic Associations in COPD Alongside NT-proBNP

**DOI:** 10.3390/diseases13100344

**Published:** 2025-10-16

**Authors:** Silvana-Elena Hojda, Teodora Mocan, Alexandra-Lucia Pop, Ramona Rusnak, Cristina Bidian, Simona Valeria Clichici

**Affiliations:** 1Department of Physiology, “Iuliu Hațieganu” University of Medicine and Pharmacy, Number 1–3, Clinicilor Street, RO-400023 Cluj-Napoca, Romania; teodora.mocan@umfcluj.ro (T.M.); cristina.bidian@umfcluj.ro (C.B.); sclichici@umfcluj.ro (S.V.C.); 2Department of Marketing, Faculty of Economics and Business Administration, Babeș-Bolyai University, Number 58–60, Teodor Mihali Street, RO-400591 Cluj-Napoca, Romania; 2203alexandra@gmail.com; 3Department of Pulmonology, City Hospital, Number 1, 22 Decembrie 1989 Street, RO-435700 Viseu de Sus, Romania; ramona_tola@yahoo.com

**Keywords:** oxidative stress, GSH, GSH/GSSG ratio, MDA, Caspase-3, NT-proBNP, LV GLS, RVFWSL, RV4CSL

## Abstract

Background/Objectives: Cardiovascular diseases are the most important cause of mortality in chronic obstructive pulmonary disease (COPD). Speckle-tracking echocardiography (2D-STE) can be used for assessing atrial and ventricular function, and its role in COPD is underexplored. The main objective of this study was to investigate prognostic associations in patients with COPD using 2D-STE echocardiography and laboratory biomarkers. Methods: The study included 70 participants, divided into two groups: 55 patients diagnosed with COPD and 15 healthy controls. All four cardiac chambers were analyzed with standard ultrasound and 2D-STE techniques. We measured NT-proBNP and several oxidative stress biomarkers: reduced glutathione (GSH), the GSH/GSSG ratio, malondialdehyde (MDA), and Caspase-3. Results: An NT-proBNP level above 325 pg/mL independently predicts advanced COPD stages (GOLD grades 3 and 4), with statistically significant results at a 95% confidence interval (CI) (*p* = 0.001). Additionally, 2D-STE identified reduced right ventricular (RV) and left ventricular (LV) strain in COPD patients before changes in LV ejection fraction. RV and LV strain measurements (RV4CLS < −16.15%, RVFWSL < −18.6%, LV GLS < −19.45%) along with PASP > 37.5 mmHg are independent predictors of advanced COPD stages, demonstrating significance at a 95% CI (*p* = 0.001). A positive correlation was observed between NT-proBNP, ultrasound parameters assessing RV systolic function, LV longitudinal strain impairment, and PASP. Conclusions: NT-proBNP serves as an independent biomarker of pulmonary hypertension and secondary right heart overload and independently predicts advanced COPD stages (GOLD grades 3 and 4) alongside RV and LV strain measurements.

## 1. Introduction

Chronic obstructive pulmonary disease (COPD) is a heterogeneous lung disease characterized by chronic respiratory symptoms, including dyspnea, cough, and airflow limitation in the airways. Risk factors include genetic or individual predisposition, lifestyle, and environmental factors, all of which favor parenchymal lung dysfunction. Exposure to cigarette smoke or other toxic particles, household or environmental pollutants, causes excessive ageing of the lung parenchyma and may contribute to COPD development [1,2]. There is a significant connection between disease severity and progression that influences the patients’ quality of life and can lead to death [3]. The last Global Initiative for Chronic Obstructive Lung Disease (GOLD) Guidelines have defined COPD as an inflammatory disorder of the respiratory system [3].

Spirometry is considered an essential method for diagnosing COPD. Spirometry screening aims to improve diagnosis, given that respiratory symptoms are a natural manifestation of smoking, and patients may neglect their symptoms and delay specialist investigations [4].

In COPD, chronic inflammation leads to airway remodeling, characterized by structural changes such as thickening of the airway wall, epithelial metaplasia, and smooth muscle hyperplasia. The mechanisms of the innate immune system promote persistent chronic inflammation in the bronchi, which leads to progressive airway remodeling and airflow obstruction during the progression of COPD [5]. The innate immune system, through the airway epithelial barrier, acts as a first line of defense against noxious particles and infections in the lung. Harmful inhaled particles, impaired immune activities, the presence of pathogens, and structural damage to the airways can trigger the release of damage-associated molecular patterns and activate innate immune cells both systemically and locally. The primary types of cells in the innate immune system that contribute to airway remodeling in COPD include inflammatory cells (neutrophils, macrophages, and lymphocytes), epithelial basal cells, dendritic cells, growth factor production, and subepithelial myofibroblast proliferation, along with the excessive production of modified matrix proteins and the thickening of all tissue layers. Airway remodeling results from the pathological processes described above and is a key factor in the progression of COPD, leading to narrowing of the airway lumen and subsequent destruction, particularly in the smaller airways [5].

Infectious exacerbations of COPD are linked to worsening clinical symptoms and influence the disease’s progression. Bacterial colonization plays a significant role in this process [6]. It is known that the bronchi commonly harbor a microbial community that helps regulate immune responses in the lungs [7]. Among respiratory bacteria, nontypeable Haemophilus influenzae (NTHi) is a major pathogen involved in colonization, acting as a primary cause of acute exacerbations and contributing to disease progression. NTHi colonizes the lungs in COPD patients, creating an environment that supports persistent infection. Platelet-activating factor (PAF) is a phospholipid mediator essential for inflammatory processes, acting on various cells such as platelets, endothelial cells, lymphocytes, and macrophages. Increased expression of the PAF receptor in the airways of COPD patients promotes bacterial adhesion and growth, aiding both acute and chronic respiratory infections in the lower respiratory tract [8]. Potential therapies aim to block the PAF receptor, showing promise in reducing pathogen adhesion and slowing COPD progression [9].

Exposure to cigarette smoke or air pollution leads to increased levels of reactive oxygen species (ROS), not only by amplifying oxidation (as indicated by markers such as GSSG, the GSH/GSSG ratio, and MDA), but also by reducing antioxidant defense mechanisms (as indicated by GSH). Oxidative stress promotes inflammation, which may lead to oxidative tissue damage in the lungs [10]. This oxidative system causes cellular dysfunction and programmed cell death (apoptosis) [11,12]. Proteases (like Caspase-3) engaged in apoptosis could serve as prognostic markers involved in preventing COPD development or worsening [13].

Brain natriuretic peptides (BNP) are produced by ventricular myocytes and released into the bloodstream due to increased myocardial stress, blood volume, or intraventricular filling pressure. NT-proBNP is the more stable form of BNP and has greater accuracy in diagnosing, managing, or risk-stratifying patients with heart failure. Recent studies have identified elevated levels of NT-proBNP in various pathologies, including COPD, where variations in plasma levels might show different disease progression stages [14,15].

## 2. Materials and Methods

### 2.1. Study Design and Characteristics of the Groups

This study involved 70 participants divided into two groups: a study group of 55 patients diagnosed with COPD from the Pulmonology and Cardiology Department of Viseu de Sus Hospital, Maramures, Romania, observed between January 2024 and January 2025, and a control group of 15 healthy non-smoker subjects of the same age as the study group. The healthy group was recruited through patient registries and physician referrals in collaboration with general practitioners (GPs). Individuals meeting the study criteria were identified and included after providing informed consent. They were healthy patients with no medical conditions, respiratory or cardiovascular symptoms, or comorbidities and were not on any ongoing treatments. The sample size calculation for the control group is detailed in the Appendix A.

The trial received the approval of the Ethics Commission of the “Iuliu Hațieganu” UMF Cluj-Napoca, Romania, under no. 222/2024.

The inclusion criteria were patients over 18 years of age in the Pulmonology or Cardiology department with a diagnosis of COPD (according to the international GOLD COPD guidelines), in GOLD grades 1–4. All patients had a history of respiratory symptoms, such as: persistent dyspnea aggravated by exertion and/or rest; chronic dry or productive cough accompanied by expectoration for at least three months; recurrent lower respiratory tract infections; in addition, a history of risk factor exposure: active or passive smoking, ex-smokers, exposure to dust, vapors, gases or other chemicals.

Patients with bronchial asthma, obstructive sleep apnea, pulmonary silicosis, idiopathic pulmonary fibrosis, a history of pulmonary thrombembolism, patients with severe left-sided heart failure with an ejection fraction < 40%, with congenital heart disease, advanced liver cirrhosis, psychiatric pathology, or recent surgery were excluded from the study. Patients in the exacerbation phase of the disease were also excluded. In addition, we excluded participants for whom we could not obtain high-quality echocardiographic images, necessary for the analysis and assessment of the measurements of interest.

The clinical examination included information such as age, gender specific predisposition, smoking/non-smoking status, index of cigarette pack/year, blood pressure (BP), heart rate (HR), duration of COPD (measured in years), number of COPD exacerbations requiring hospitalization within one year, and constant self-therapy administered at home. The pack-year index was calculated using the formula (number of cigarettes used daily x smoking duration in years). We collected all the data on each study participant’s medical history and associated comorbidities.

### 2.2. Laboratory Parameter Analysis

The blood samples were collected under aseptic conditions, and all tests were done by professional technicians in the Physiology Department of UMF “Iuliu Hațieganu”. The complete blood count values were assessed through fluorescence flow cytometry (Myndray BC-5380, Roche, Penzberg, Germany). The C-reactive protein (CRP) was analyzed using latex-enhanced immunoturbidimetry (Vitros 5600, Ortho Clinical Diagnostics, Raritan, NJ, USA; range 0–5 mg/L). After centrifugation at 1000× *g* for 15 min, the supernatant obtained from the patient’s serum was frozen at −80 °C until all study participants had their samples harvested. NT-proBNP levels were measured using a human ELISA kit (Human NT-proBNP ELISA Kit, MCT, Sigma-Aldrich, Bournemouth, UK, detection range: 15.9–1000 pg/mL, sensitivity: 14 pg/mL). Oxidative stress parameters with potential implications in COPD were calculated using the following methods: lipid peroxidation by measuring malondialdehyde (MDA) using the colorimetric Conti method [16] (detection range: 0.5–40 nmol/mL, sensitivity 0.24 nmol/mL), reduced glutathione (GSH) using the colorimetric Hu method [17] (detection range: 0.36–30 nmol/mL, sensitivity 0.28 nmol/mL), and oxidized glutathione (GSSG) using the Vats method [18] (detection range: 0.32–15 nmol/mL, sensitivity 0.15 nmol/mL). The GSH/GSSG ratio was used to assess antioxidant activity and was calculated as a marker of oxidative stress. Caspase-3 was measured in patient serum using a human ELISA kit (Human Caspase-3 ELISA, Blue Gene Biotech, Shanghai, China) with a detection range of 78.1–5000 pg/mL and a sensitivity of 46.88 pg/mL.

### 2.3. Echocardiographic Assessment

All images were acquired using a Vivid E95 ultrasound machine (GE Vingmed Ultrasound, Horten, Norway). For each echocardiogram performed in expiratory apnea, at least three cardiac cycles were recorded and stored in a DICOM database. These were later analyzed offline with EchoPac BT13 software (GE Vingmed Ultrasound, Norway), using modern two-dimensional speckle-tracking echocardiography (2D-STE) techniques, following the European Society of Echocardiography recommendations [19,20].

The following measurements were used to assess RV systolic function: the tricuspid annular plane systolic excursion (TAPSE) by placing the M-mode cursor at the junction between the tricuspid valve plane and the RV free wall, systolic S’ wave by placing the pulsed-wave tissue Doppler sample at the level of the lateral tricuspid annulus, RV area percent change (FAC-fractional area change), and using current 2D-STE techniques web evaluated the RV free wall longitudinal strain (RVFWSL) and the RV four-chamber longitudinal strain (RV4CSL) [20,21]. To assess left ventricle (LV) systolic function, the following parameters were used: LV ejection fraction (LVEF) and, using current 2D-STE techniques, LV global longitudinal strain (LV GLS).

The following measurements were also taken: the left atrial (LA) volume, indexed to body surface area, was calculated. Using current 2D-STE techniques, LA strain was analyzed: LASr-left atrial reservoir strain, LAScd-left atrial conduit strain, and LASct-left atrial contractile strain, as described in [22,23]. The right atrial (RA) area and volume were also analyzed [24]. Pulmonary artery systolic pressure (PASP-mmHg) was calculated from the pressure gradient between the RV-RA and the pressure in the RA, according to Bernoulli’s equation [25]. All values obtained were compared to the normal values provided by the EACVI guidelines [25,26].

### 2.4. Spirometry

Spirometric measurements were performed using a SPIRO PRO Plus BTL-08 (UK), following European COPD Guidelines recommendations [3]. According to the GOLD 2025 criteria, a COPD diagnosis was confirmed when the forced expiratory volume in one second (FEV1%) relative to the forced vital capacity (FVC) was less than 70% (FEV1/FVC < 0.7) [3]. 

The graphical representation of the study design can be seen in Figure 1.

### 2.5. Statistical Analysis

Descriptive and inferential analysis methods were used to process the collected data. Thus, contingency tables (for qualitative variables) and dispersion and central tendency indices (continuous numerical variables) were used to describe data. Chi-square or Fisher’s Exact Tests were used to compare differences between groups in the case of qualitative data. Continuous variables were tested for normality using the Kolmogorov–Smirnov test, with subsequent choice of the use of tests depending on the result returned by each variable. We used the Student or Mann–Whitney U Test to compare different groups, as appropriate. Correlations between variables were calculated using the Spearman R regression index. To evaluate the ability of each biomarker or echocardiographic parameter to discriminate mild/moderate stages of COPD (GOLD grades 1 and 2) from severe/very severe stages of COPD (GOLD grades 3 and 4), ROC analysis was used to assess their sensitivity and specificity. The ROC analysis provided an AUC (area under the curve) metric for the echocardiographic measurements and biomarkers, 95% CI (confidence interval) for AUC, and an optimal cut-off value estimated based on the Youden index. Based on the optimal cut-off estimated values, we also tested a univariate logistic binomial model, a multivariate logistic binomial model, and linear regression to assess if the biomarkers or echocardiographic measurements remained significant predictors for severe/very severe stages of COPD. For all tests, the significance threshold of *p* ≤ 0.05 was chosen. The statistical package IBM SPSS Statistics 20 was used for data processing.

## 3. Results

Only results that are statistically significant and relevant are highlighted below. Additional information concerning demographic data, anthropometric measurements, spirometry results, comorbidities, and medication use of participants in the control and COPD groups was provided in the Appendix A.

The results of the statistical analysis of the ultrasound measurements in the two groups are shown below. Table 1 and Table 2 highlight the diversity of echocardiographic measurement values assessed in the control group and the group of patients diagnosed with COPD, according to GOLD severity grades (1–4).

A significant degree of global longitudinal dysfunction of the LV (GLS LV), and clinical and subclinical impairment of RV systolic function (S’ RV, TAPSE, FAC-RV, RVFWSL, RV4CSL, and PASP) were observed between the two studied groups and within the group of COPD patients, according to GOLD severity grades.

Table 2 highlights the diversity of echocardiographic measurement values assessed in all four GOLD severity grades.

Regarding laboratory parameters, Table 3 shows the significant differences observed between the control group and patients diagnosed with COPD. The extended statistical analysis is available in Appendix A. Inflammatory markers (CRP, NLR), oxidative stress markers (MDA, GSH, GSSG, GSH/GSSG ratio, Casp 3), and NT-proBNP exhibit increased values in the COPD group.

Table 4 shows significant differences in paraclinical parameter values and spirometry results based on COPD severity, as classified by GOLD grades and airflow obstruction severity. The complete statistical analysis of laboratory parameters, grouped by GOLD severity grades (1–4), can be found in Appendix A.

Table 5 highlights the lowest, highest, and average values of NT-proBNP in all four stages of COPD severity. According to the latest PH guidelines, an NT-proBNP value < 300 pg/mL predicts a low probability of PH [26].

The frequency of COPD acute exacerbations (AECOPD) per year showed significant differences in Table 6. There was a directly proportional increase in inflammatory markers (CRP, NLR) and NT-proBNP levels among patients with one or three exacerbations. Caspase-3 levels were significantly higher in patients with three exacerbations per year compared to those with only one, among oxidative stress markers. A multiple linear regression model demonstrated that for every one-unit increase in AECOPD, Caspase-3 increases by 50.2 pg/mL (*p =* 0.009). Significant global longitudinal dysfunction of the LV (GLS LV), and clinical and subclinical impairment of RV systolic function (S’ RV, TAPSE, FAC-RV, RVFWSL, RV4CSL, and PASP) were observed in patients with three exacerbations per year compared to those with only one. A multiple linear regression model showed that impaired RV systolic function was directly related to the frequency of exacerbations. For each additional AECOPD, the TAPSE value decreases by 1.11 mm, the S’ RV value decreases by 0.94 mm, and the RVFWSL value increases by 1.25%. An increase in pulmonary hypertension (PH) was also associated with higher exacerbation frequency, with the PASP value increasing by 2.5 mmHg for each additional AECOPD.

Using the Matrix of Spearman’s rank correlations in Table 7 can be observed that elevated serum NT-proBNP levels are significantly correlated with echocardiographic measurements and spirometry values. There is also a lack of significant correlations between the measured oxidative stress markers and the ultrasound or spirometry measurements.

As shown in Figure 2, the receiver operating characteristic (ROC) analysis indicated an estimated area under the curve for NT-proBNP levels predicting severe/very severe stages of COPD of 0.88 (95% CI: 0.78–0.98, *p* = 0.001 < 0.05) and an optimal cut-off value based on the Youden index of 325 pg/mL, with a sensitivity (Se) of 52% and a specificity (Sp) of 88%.

The performance of the CRP level in differentiating patients with severe or very severe COPD from those with mild or moderate COPD was as follows: AUC = 0.775 (95% CI: 0.64–0.90, *p* = 0.001 < 0.05), with an optimal cut-off of 12.25 mg/L, showing a sensitivity of 95.7% and a specificity of 71.9%.

The markers of oxidative stress (MDA, GSH, GSSG, GSH/GSSG ratio, Caspase-3) did not show significant results.

In Table 8, univariate and multivariate logistic models indicated that, after adjusting for other risk factors, CRP < 12.25 mg/L versus CRP > 12.25 mg/L and NT-proBNP < 325 pg/mL versus NT-proBNP > 325 pg/mL were independent predictors of either mild/moderate COPD stages (GOLD grades 1 and 2) or severe/very severe stages (GOLD grades 3 and 4). As shown in Model 3, CRP and NT-proBNP directly and positively affect the severity stages of COPD.

As shown in Figure 3, the ROC analysis indicated an estimated AUC for LV GLS predicting severe or very severe stages of COPD of 0.925 (95% CI: 0.850–1.0, *p* = 0.001 < 0.05), with an optimal cut-off value of −19.45%, Sp = 59.40%, Se = 95.70%. LV GLS had high specificity but low sensitivity. Additionally, RV4CLS and RVFWSL predicted severe or very severe stages of COPD with AUCs of 0.988 (95% CI: 0.969–1.0, *p* = 0.001 < 0.05) and 0.995 (95% CI: 0.983–1.0, *p* = 0.001 < 0.05), respectively. The optimal cut-off value for RV4CLS was −16.15%, with Sp = 9.4% and Se = 95.7%, while for RVFWSL it was −18.6%, with Se = 91.30% and Sp = 3.1%.

RV4CLS and RVFWSL had high sensitivity but low specificity.

PASP, according to ROC analysis, predicts severe/very severe stages of COPD with an AUC of 0.804 (95% CI: 0.691–0.918, *p* = 0.001 < 0.05), an optimal cutoff value of 37.50 mmHg, with a Sp of 46.9% and a Se of 87%. PASP showed high sensitivity but low specificity.

In Table 9, univariate and multivariate logistic models showed that, after adjusting for other risk factors, LV GLS < −19.45% versus > −19.45%, RV4CLS < −16.15% vs. > −16.15%, and RVFWSL < −18.6% versus > −18.6% were independent predictors of severe or very severe COPD stages (GOLD grades 3 and 4) compared to mild or moderate stages (GOLD grades 1 and 2).

## 4. Discussion

This study involved patients diagnosed with COPD and healthy participants. Most participants were in their 60 s and 70 s, a similar age range to that of other clinical trials assessing COPD severity, which correlated with echocardiographic and laboratory parameters [30,31]. According to statistical analysis, there were no significant differences in general characteristics between the two groups. 

Regarding laboratory parameters, NT-proBNP levels were significantly higher in patients diagnosed with COPD. Furthermore, this clinical study revealed significant differences in plasma NT-proBNP levels depending on COPD severity. NT-proBNP exhibited high specificity when distinguishing between patients with severe or very severe COPD and those with mild or moderate disease severity, with an optimal cut-off value of 325 pg/mL. This finding confirmed that NT-proBNP could serve as an independent prognostic biomarker for advanced stages of COPD when levels exceed 325 pg/mL. Regarding the frequency of exacerbations per year, the study showed that patients experiencing more than three exacerbations annually had elevated NT-proBNP levels, as well. Serum NT-proBNP levels were significantly negatively correlated with FEV1% or FVC obtained by spirometry. According to a recent meta-analysis, these results demonstrate again that NT-proBNP is directly related to the severity of airway obstruction, particularly in patients with FEV1 < 50% [15]. Hawkins et al. described elevated serum levels of NT-proBNP in patients with COPD during AECOPD compared to those with stable disease [32]. In over six clinical trials, increased serum levels of NT-proBNP were directly related to disease severity in patients with FEV1 < 50% [15]. This was also confirmed in the present study.

In addition, the literature suggests that increased levels of natriuretic peptides may also be influenced by the pro-inflammatory activity associated with this pathology [32]. Our research also confirmed this aspect, as a significant correlation was observed between plasma levels of NT-proBNP, CRP, and NLR.

We would also like to point out that there is a positive correlation between the NT-proBNP value and the ultrasound parameters assessing RV systolic function (S’ RV, TAPSE, RV FAC, RVFWSL, RV4CSL), as well as LV longitudinal strain impairment (LV GLS) and PASP. An NT-proBNP value < 300 pg/mL predicts a low probability of PH, according to the latest European Guidelines [26].

Clinical studies have shown that in patients diagnosed with COPD, both pulmonary vascular pressure and right heart preload increased secretion of NT-proBNP [15]. Even though patients with severe cardiovascular pathology (those with heart failure with reduced ejection fraction < 40%) were excluded from this study, there still appears to be a pulmonary vascular remodeling implication. Elevated NT-proBNP in COPD often reflects RV dysfunction, PH occurrence, increased RV preload, RV failure, and increased traction forces on the RV walls rather than being a direct indicator of airflow limitation stage. Our study also highlighted this aspect.

Based on the data obtained from this study and the literature [33], NT-proBNP can be used for predicting advanced stages of COPD and as an adjuvant in identifying inflammatory status and cardiovascular stress associated with COPD.

Moreover, this clinical study revealed that both RV4CLS and RVFWSL had high sensitivity when distinguishing patients with severe/very severe stages of COPD from those with mild or moderate disease severity at an optimal cut-off value of RV4CLS < −16.15% and RVFWSL < −18.6%. RV longitudinal strain measurement assessed using 2D-STE is a promising, non-invasive method for evaluating subclinical RV dysfunction. It serves as a relevant prognostic marker in lung disease, according to the literature [34,35]. For each increase in AECOPD by one unit, the TAPSE value decreases by 1.11 mm, the S’RV decreases by 0.94 mm, and the RVFWSL value increases by 1.25%.

PH was significantly associated with COPD severity and with the increase in AECOPD per year. A PASP value > 37.50 mmHg had high sensitivity when distinguishing between patients with severe or very severe COPD and those with mild or moderate disease severity. Similarly, a one-unit increase in AECOPD corresponds and an increase of 2.5 mmHg in PSAP. In fact, 65% of the patients in this study developed PH with PASP exceeding 37 mmHg. These pathological changes in the right heart structures are already known and have been described in the literature. Pulmonary vascular remodeling and pulmonary vasoconstriction cause increased RV preload and RV free wall hypertrophy in the early stages of COPD, followed by RV dilatation, RV systolic dysfunction, PH, right heart failure, and chronic cor in severe and very severe stages of the disease [26,30].

It is important to note that any impairment of right heart function also requires a comprehensive evaluation of the left side of the heart, as demonstrated by studies published so far [36,37]. This study found no significant differences in the LVEF value between the GOLD grades of COPD and the control groups. The study included 10% of patients with LVEF between 40–50%, but most had LVEF greater than 50% (90%), although LVEF remained approximately the same in the two groups of interest. Longitudinal dysfunction of LV systolic function was assessed using 2D speckle-tracking techniques in patients with EF over 50%. The results align with the existing literature [35,36,37,38]. LV GLS showed high sensitivity with an optimal cut-off value of GLS < −19.45% for distinguishing patients with severe or very severe COPD from those with mild or moderate disease, consistent with the literature data [23]. This study aims to reaffirm existing findings indicating that subclinical assessment of LV systolic function using GLS is essential in this condition. It provides insight into subtle changes in LV contractility related to disease severity and progression.

The use of these measurements for assessing subclinical RV and LV dysfunction (of RV4CLS < −16.15%, respectively, RVFWSL < −18.6%, LV GLS < −19.45%) should be employed for COPD patients, especially in situations where RV and LV systolic function, as assessed by conventional measurements, remains within normal limits. These three echocardiographic parameters may be considered relevant prognostic measurements in patients with COPD, along with a PASP > 37.5 mmHg.

However, in this study, the value of the ultrasound parameters obtained from the evaluation of the LA and RA did not differ between the group of patients diagnosed with COPD and the control group. In addition, the advanced speckle-tracking techniques of LA function assessment (LASr, LAScd, and LAScd) did not show significant differences between the two studied groups. Over time, the LA strain has gained more and more attention in clinical practice as an important tool for assessing atrial function [31]. A recently published study by Dang et al., conducted on 780 patients with COPD, obtained the same results as the present study [22].

In this study, serum CRP levels have high sensitivity and specificity in discriminating between patients with severe/very severe stages of COPD and those with mild/moderate stages of COPD, at an optimal cut-off value of 12.25 mg/L. This observation reconfirmed that CRP is an independent predictor for the advanced form of COPD and is the most used biomarker for confirming AECOPD; this aspect is already known in the literature [39,40].

Additionally, we aimed to analyze various oxidative stress markers, including GSH, the GSH/GSSG ratio, Caspase-3, and MDA [41,42,43]. However, we only observed a significant increase in these biomarkers in the group of patients with COPD compared to the control group. This study highlighted an increase in the frequency of AECOPD, with serum Caspase-3 levels rising by 50.2 pg/mL for each AECOPD. Therefore, Caspase-3 could serve as a prognostic marker of cell apoptosis severity in COPD patients and as a predictor of infectious exacerbations. Caspase-3 inhibition shows potential for treating COPD by preventing structural cell apoptosis, as seen in animal models where caspase-3 inhibitors reduced lung damage [44]. More information about those oxidative stress markers measured in this study can be found in the Appendix A.

Over time, concerns have been raised about the reliability of spirometry. Schermer et al. reported a small but statistically significant difference between lung function values obtained in a doctor’s office and those measured in a pulmonary function laboratory [45]. A persistent issue remains with the inconsistent use and interpretation of spirometry results by physicians, despite scientific societies emphasizing that COPD diagnosis and treatment should only begin after objective spirometry testing and interpretation by specialized clinicians. A significant number of COPD diagnoses in primary care are false positives, and the use of spirometry to confirm the diagnosis is limited [46,47]. Other associations with prognostic value could be used in assessing COPD severity. NT-proBNP measurement is one such biomarker. Advanced 2D-ST techniques could also be utilized as a valuable tool in evaluating RV and LV systolic function in COPD prognosis. There is a need for larger future studies in this direction.

The limitations of this study include a small number of patients from a single hospital in Romania. Due to the limited number of patients in the GOLD 1 grade COPD, the statistical analysis is unreliable because of the small sample size. It was not possible to provide the necessary information to determine whether subclinical ventricular dysfunction of the LV or RV, or the presence of PH, could be identified in the early stages of the disease. Conducting multicenter studies in different locations worldwide would be necessary to verify the results reliably. We obtained serum biomarker levels only at the time of admission and had no data on changes in their serum concentrations during readmissions for acute exacerbations of worsening COPD. Another significant limitation is the quality of the cardiac ultrasound images. Additionally, there are anatomical challenges, as multiple sections are needed to obtain accurate measurements.

## 5. Conclusions

NT-proBNP, a biomarker frequently used in clinical practice for cardiovascular disease assessment, has proven to be an effective indicator for evaluating cardiac dysfunction or pulmonary hypertension in patients with COPD. Levels of NT-proBNP above 325 pg/mL can indicate advanced stages of COPD, pulmonary hypertension, and secondary right heart overload independently. When combined with echocardiographic measurements of biventricular longitudinal systolic dysfunction, they may serve as independent predictors of advanced COPD stages.

## Figures and Tables

**Figure 1 diseases-13-00344-f001:**
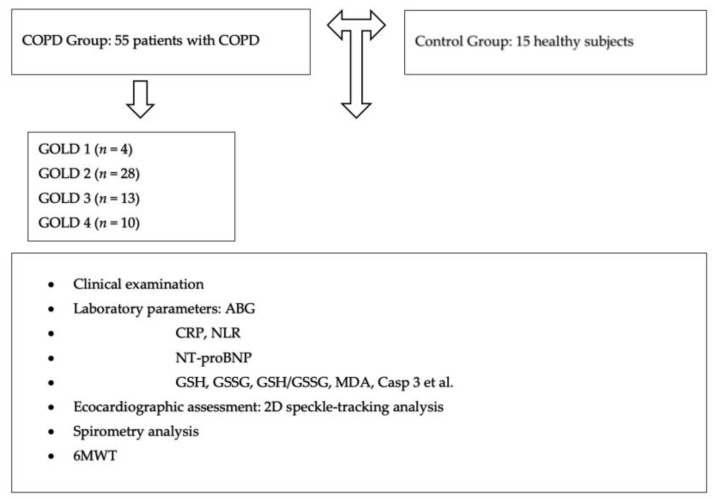
Flowchart of assessment of chronic obstructive pulmonary disease (COPD) patients and healthy participants.

**Figure 2 diseases-13-00344-f002:**
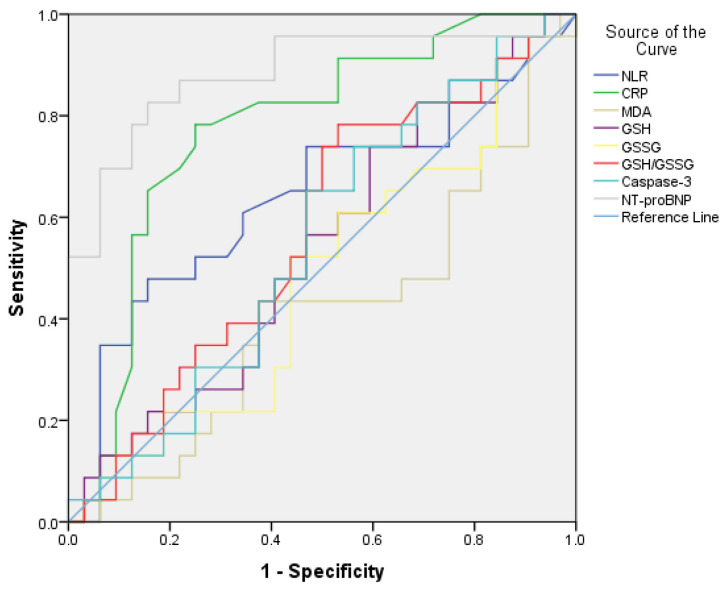
ROC plot of serum levels of CRP and NT-proBNP for predicting the presence of severe/very severe forms of COPD.

**Figure 3 diseases-13-00344-f003:**
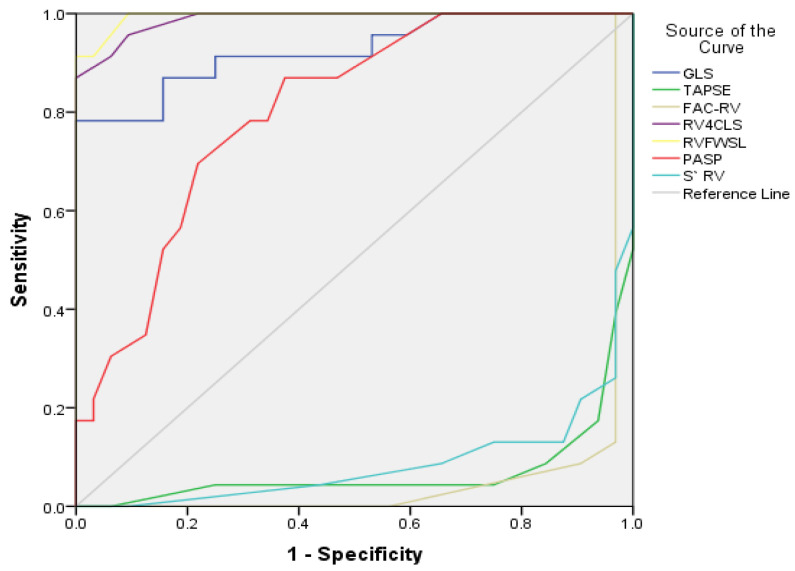
ROC plot of echocardiographic measurements for predicting the presence of severe/very severe stages of COPD (GOLD grades 3 and 4).

**Table 1 diseases-13-00344-t001:** Echocardiographic measurement values in the control group and the COPD group [23,25,27].

Variables/Normal Values	COPD Group (*n* = 55)	Control Group (*n* = 15)	*p*-Values
LVEF (%)/>50%	55 ± 4.3	57 ± 3.6	0.43
GLS LV avg (%)/>18%	−19.9 ± 1.93	−20.9 ± 0.9	<0.001 *
S’ RV (cm/s)/>10 cm/s	12.0 ± 2.7	13.6 ± 1.5	0.03 *
TAPSE (mm)/>16 mm	18.8 ± 3.1	23.3 ± 1.5	<0.001 *
FAC-RV (%)/>35%	32.1 ± 9.8	36.7 ± 2.1	<0.001 *
RVFWSL%/–26.7% ± 5.2	−18.3 ± 1.7	−23.0 ± 1.2	<0.001 *
RV4CSL%/–21.7% ± 3.4%	−15.9 ± 1.8	−23.0 ± 1.2	<0.001 *
RA aria (cm^2^)/>18 cm^2^	17.3 ± 2.1	15.4 ± 2.4	0.06
RA volume (mL/m^2^)/>31 mL/m^2^	29.1 ± 5.4	27.2 ± 3.4	0.07
LA volume (mL/m^2^)/<34 mL/m^2^	33.9 ± 4.9	32.4 ± 1.5	0.09
LASr (%)/35.9% ± 10.6%	31.4 ± 6.9	37.9 ± 5.4	0.08
LAScd (%)/−21.9% ± 9.3%	−21.1 ± 5.7	−24.1 ± 6.8	0.06
LASct (%)−13.9% ± 3.6%	−16.8 ± 3.2	−17.1 ± 1.8	0.09
PASP (mmHg)/15–30 mmHg	34.8 ± 5.7	22.3 ± 4.3	<0.001 *

The results were expressed as mean ± SD (standard deviation). LVEF—left ventricular ejection fraction, LV GLS avg—average global longitudinal strain of the LV, RV—right ventricle, S’RV—systolic longitudinal contraction of the RV, TAPSE—systolic motion of the tricuspid annulus plane, FAC—fractional aria change, RVFWSL—RV free wall longitudinal strain, RV4CSL—RV four chamber longitudinal strain, RA—right atrium, LA—left atrium, LASr—reservoir function of the LA, LAScd—conduit function of the LA, LASct—contraction function of the LA, PASP—pulmonary artery systolic pressure; * significant results: *p*-values were obtained by Pearson Correlation, * *p* < 0.05.

**Table 2 diseases-13-00344-t002:** Echocardiographic measurement values according to GOLD grades (1–4) [23,25,27].

Variables/Normal Values	COPD GOLD 1	COPD GOLD 2	COPD GOLD 3	COPD GOLD 4	*p*-Values
LVEF (%)/>50%	56.7 ± 2.5	56.6 ± 2.7	53 ± 4.9	51.5 ± 4.5	<0.06
GLS LV avg (%)/>18%	−21.3 ± 0.8	−20.1 ± 1	−17.9 ± 1.4	−16.6 ± 1.2	<0.001 *
S’ RV (cm/s)/>10 cm/s	15.2 ± 0.5	13.4 ± 1.8	9.8 ± 1.4	9.4 ± 2.5	<0.001 *
TAPSE (mm)/>16 mm	21.2 ± 0.9	20.7 ± 2.1	17.3 ± 2	14.5 ± 0.9	<0.001 *
FAC-RV (%)/>35%	43.7 ± 2.2	37.1 ± 7.2	29.3 ± 4.3	17.3 ± 2	<0.001 *
RVFWSL%/–26.7% ± 5.2	−21 ± 0.3	−19.3 ± 0.4	−17.5 ± 0.9	−15.4 ± 0.7	<0.001 *
RV4CSL%/–21.7% ± 3.4%	−18.7 ± 0.4	−17 ± 0.4	−15.2 ± 0.9	−13 ± 0.7	<0.001 *
RA aria (cm^2^)/>18 cm^2^	15.5 ± 1.2	17.5 ± 2.1	17.3 ± 1.8	17.6 ± 2.3	0.31
RA volume (mL/m^2^)/>31 mL/m^2^	24.7 ± 1.7	29 ± 4.9	30.3 ± 6.5	29.5 ± 5.9	0.37
LA volume (mL/m^2^)/<34 mL/m^2^	31.2 ± 2	34.1 ± 4.4	34.1 ± 6.5	34.1 ± 5	0.75
LASr (%)/35.9% ± 10.6%	31 ± 6.7	30.2 ± 5.2	28 ± 6.6	28.7 ± 4.5	0.07
LAScd (%)/−21.9% ± 9.3%	−22.2 ± 2.8	−22.4 ± 4.5	−21 ± 6.6	−19.8 ± 6.4	0.06
LASct (%)−13.9% ± 3.6%	−17.7 ± 1.8	−17.3 ± 3.3	−15.3 ± 2.1	−16.2 ± 3.7	0.08
PASP (mmHg)/15–30 mmHg	27.7 ± 2	33 ± 4.9	35.4 ± 4.3	42 ± 3.5	<0.001 *

The results were expressed as mean ± SD (standard deviation). LVEF—left ventricular ejection fraction, GLS LV avg—average global longitudinal strain of the LV, RV—right ventricle, S’ RV—systolic longitudinal contraction of the RV, TAPSE—systolic motion of the tricuspid annulus plane, FAC—fractional aria change, RVFWSL—RV free wall longitudinal strain, RV4CSL—RV four chamber longitudinal strain, RA—right atrium, LA—left atrium, LASr—reservoir function of the LA, LAScd—conduit function of the LA, LASct—contraction function of the LA, PASP—pulmonary artery systolic pressure; * significant results: *p*-values were obtained by Pearson Correlation, * *p* < 0.05.

**Table 3 diseases-13-00344-t003:** Significant differences in laboratory parameter values between the control group and the COPD group [16,17,18,28].

Variables/Normal Values	COPD Group (*n* = 55)	Control Group (*n* = 15)	*p*-Values
NLR (%)/0.78–3.53%	4.46 ± 3.1	2.62 ± 0.5	0.030 *
CRP (mg/L)/0–5 mg/L	22.1 ± 24.9	3 ± 0.7	0.004 *
MDA (nmol/mL)/0.5–40 nmol/mL	2952.4 ± 948.3	1442.0 ± 959.5	0.001 *
GSH (nmol/mL)/0.36–30 nmol/mL	6.9 ± 2.0	8.6 ± 2.4	0.008 *
GSSG (nmol/mL)/0.32–15 nmol/mL	1.2 ± 0.50	0.7 ± 0.3	0.004 *
GSH/GSSG	5.9 ± 2.7	14.6 ± 8.65	0.001 *
Casp 3 (pg/mL)/78.13–5000 pg/mL	240.8 ± 119.2	170.1 ± 75.0	0.03 *
NT-proBNP (pg/mL)/<300 pg/mL	480.0 ± 471.6	78.3 ± 17.9	0.002 *

The results were expressed as mean ± SD (standard deviation). NLR-neutrophil/lymphocyte ratio, CRP-C-reactive protein, MDA-malondialdehyde, GSH-reduced glutathione, GSSG-oxidized glutathione, GSH/GSSG-reduced glutathione/oxidized glutathione ratio, Casp 3—Caspase-3, NT-proBNP-N-terminal pro-Brain Natriuretic Peptide; * significant results; *p*-values were obtained by Pearson Correlation, * *p* < 0.05.

**Table 4 diseases-13-00344-t004:** Paraclinical parameter values based on GOLD grades (1–4) [3,26].

Variables/Normal Values	COPD GOLD 1	COPD GOLD 2	COPD GOLD 3	COPD GOLD 4	*p*-Values
Laboratory					
NLR (%)/0.78–3.53%	2.1 ± 2	4.1 ± 3	4.3 ± 2.8	6.6 ± 3.4	0.05 *
CRP (mg/L)/0–5 mg/L	13.7 ± 20	21.1 ± 16	26.3 ± 24.9	45.2 ± 27.2	0.005 *
NT-proBNP (pg/mL)/<300 pg/mL	148.5 ± 39.5	301.2 ± 111.3	598.6 ± 685.8	958.9 ± 496.2	<0.001 *
Spirometry					
FEV 1%/>80% [3]	85 ± 3.3	65.7 ± 8.5	45.2 ± 3.4	27 ± 7.3	<0.001 *
FVC (L)/3.2–4.5 L [3]	4.3 ± 0.1	3.6 ± 0.2	3.1 ± 0.1	2.6 ± 0.1	<0.001 *

Results were expressed as mean ± SD (standard deviation). NLR—neutrophil/lymphocyte ratio, CRP—C-reactive protein, NT-proBNP-N-terminal pro-Brain Natriuretic Peptide, FEV1-forced expiratory volume in one second, FVC-forced vital capacity; * significant results: *p*-values were obtained by Pearson Correlation, * *p* < 0.05.

**Table 5 diseases-13-00344-t005:** The lowest, highest, and average values of NT-proBNP in GOLD grades (1–4) [26].

	Number of Patients	Minimum	Maximum	Mean	SD
COPD GOLD 1	4	109.00	196.00	148.50	39.53
COPD GOLD 2	28	117.00	580.00	301.29	111.35
COPD GOLD 3	13	103.00	2830.00	598.69	685.89
COPD GOLD 4	10	632.00	2300.00	958.90	496.29

The results were expressed as maxim, minim, mean and SD (standard deviation).

**Table 6 diseases-13-00344-t006:** Statistically significant values of laboratory parameters, spirometry values, and ultrasound measurements depending on the severity of COPD exacerbations/year [3,23,25,26,27,29].

Variables	1 Exacerbation/Year	2 Exacerbations/Year	3 Exacerbations/Year	*p*-Values
Laboratory parameter values				
NLR (%)/0.78–3.53%	3.2 ± 2.1	4.1 ± 2.8	6.4 ± 3.7	0.006 *
CRP (mg/L)/0–5 mg/L	16.6 ± 21.1	29.2 ± 3.3	35.9 ± 1.8	0.007 *
Casp 3 (pg/mL)/78.13–5000 pg/mL	180.4.6 ± 37.6	279.6 ± 142.1	276.3 ± 134.5	0.01 *
NT-proBNP (pg/mL)/<300 pg/mL	332.1 ± 183.6	313.8 ± 148.2	861 ± 715.8	<0.001 *
Spirometry values				
FEV 1%/>80% [3]	63.3 ± 19.1	60.2 ± 10.3	38.8 ± 14.6	<0.001 *
FVC (L)/3.2–4.5 L [3]	3.6 ± 0.4	3.5 ± 0.3	2.9 ± 0.4	<0.001 *
Echocardiography parameter values				
LVEF (%)/>50%	56.8 ± 2.8	55.5 ± 3.5	52 ± 5.2	0.05
GLS LV avg (%)/>18%	−19.9.3 ± 1.6	−19.5 ± 1.4	−17.4 ± 1.8	<0.001 *
S’ RV (cm/s)/>10 cm/sec	12.3 ± 2.5	13.1 ± 2.1	10.2 ± 2.8	0.005 *
TAPSE (mm)/>16 mm	19.3 ± 2.6	20.2 ± 2.5	16.8 ± 3.5	0.004 *
FAC-RV (%)/>35%	34.4 ± 10	36.7 ± 5.3	24 ± 8.7	<0.001 *
RVFWSL %/–26.7% ± 5.2	−19.2 ± 1.2	−18.8 ± 1.1	−16.6 ± 1.8	<0.001 *
RV4CSL %/–21.7% ± 3.4%	−16.8 ± 1.3	−16.5 ± 1	−14.2 ± 1.8	<0.001 *
PASP (mmHg)/15–30 mmHg	33.6 ± 5.3	32.6 ± 4.7	38.8 ± 5.6	<0.001 *

The results were expressed as mean ± SD (standard deviation). NLR-neutrophil/lymphocyte ratio, CRP—C-reactive protein, Casp 3—Caspase-3, NT-proBNP-N-terminal pro-Brain Natriuretic Peptide, FEV1—forced expiratory volume in one second, FVC-forced vital capacity, NLR—neutrophil/lymphocyte ratio, CRP—C-reactive protein, LVEF—left ventricular ejection fraction, GLS LV avg—average global longitudinal strain of the LV, RV—right ventricle, S’ RV—systolic longitudinal contraction of the RV, TAPSE—systolic motion of the tricuspid annulus plane, FAC—fractional aria change, RVFWSL—RV free wall longitudinal strain, RV4CSL—RV four chamber longitudinal strain, PASP—pulmonary artery systolic pressure; * significant results: *p*-values were obtained by Pearson Correlation, * *p* < 0.05.

**Table 7 diseases-13-00344-t007:** Matrix of Spearman’s rank correlations between MDA, GSH, GSSG, GSH/GSSG, Caspase-3, NT-proBNP and clinical characteristics, laboratory parameters, spirometry, and echocardiographic measurements.

Variables	GSH	GSSG	GSH/GSSG	Casp 3	NT-proBNP
Smoking index (pack/year)	−0.07; 0.56	0.16; 0.21	−0.27; 0.03 *	0.15; 0.26	0.23; 0.51
Age, years	−0.27; 0.05 *	0.2; 0.13	−0.37; 0.005 *	0.32; 0.01 ^*^	0.17; 0.19
FEV1 (%)	−0.08; 0.55	−0.04; 0.74	−0.10; 0.42	−0.08; 0.54	−0.68; 0.001 *
FVC (L)	0.27; 0.05 *	0.08; 0.53	0.27; 0.04 *	−0.02; 0.84	−0.65; 0.001 *
LV GLS (%)	0.11; 0.39	−0.06; 0.63	0.24; 0.06	0.13; 0.32	−0.54; 0.001 *
S’ RV (cm/s)	−0.02; 0.86	0.01; 0.91	−0.08; 0.55	−0.08; 0.54	−0.47; 0.001 *
TAPSE (mm)	−0.15; 0.24	−0.10; 0.44	−0.07; 0.59	−0.05; 0.71	−0.53; 0.001 *
FAC-RV (%)	−0.01; 0.92	−0.02; 0.83	−0.13; 0.34	−0.21; 0.11	−0.75; 0.001 *
RVFWSL %	0.19; 0.15	0.01; 0.90	0.22; 0.09	0.15; 0.27	−0.66; 0.001 *
RV4CSL %	0.22; 0.09	0.01; 0.97	0.26; 0.05	0.13; 0.3	−0.66; 0.001 *
PASP (mmHg)	−0.03; 0.82	−0.19; 0.15	0.16; 0.24	0.02; 0.84	0.65; 0.001 *

The results were expressed as mean ± SD (standard deviation). *p*-value, *—statistically significant results, *p* < 0.05, GSH—reduced glutathione, GSSG—oxidized glutathione, GSH/GSSG—reduced glutathione/oxidized glutathione ratio, Casp 3—Caspase-3, FEV1—forced expiratory volume in the first second, FVC—forced vital capacity, LV GLS—global longitudinal strain of the LV, RV—right ventricle, S’ RV—systolic longitudinal contraction of the RV, TAPSE—systolic motion of the tricuspid annulus plane, FAC—fractional area change, RVFWSL-RV free wall longitudinal strain, RV4CSL—RV four chamber longitudinal strain, PASP—pulmonary artery systolic pressure.

**Table 8 diseases-13-00344-t008:** Association between CRP and NT-proBNP and the odds of severe/very severe (GOLD grades 3 and 4) or mild/moderate (GOLD grades 1 and 2) stages of COPD in all tested models.

Severity Stages of COPD	Variables	[95% CI]	*p*-Value
Model 1			
Mild/moderate stages of COPD (GOLD grades 1 and 2) Severe/very severe stages of COPD (GOLD grades 3 and 4)	CRP NT-proBNP	[3.241–29.325] [246.965–699.269]	0.015 * 0.000 *
Model 2			
Mild/moderate stages of COPD (GOLD grades 1 and 2) Severe/very severe stages of COPD (GOLD grades 3 and 4)	CRP NT-proBNP	[3.821–30.112] [276.313–714.513]	0.012 * 0.000 *
Model 3			
Mild/moderate stages of COPD (GOLD grades 1 and 2) Severe/very severe stages of COPD (GOLD grades 3 and 4)	CRP NT-proBNP	Y (severity stages of COPD) = −0.962 + 0.028 × CRP Y (severity stages of COPD) = −4.125 + 0.009 × NT-proBNP	0.024 * 0.000 *

Model 1: univariate model. Model 2: multivariate model including the biomarker and age, sex, and active smoking as covariates. Model 3: binomial logistic regression model, CI = 95% confidence interval; * significant result: *p* < 0.05.

**Table 9 diseases-13-00344-t009:** Relationship between echocardiographic measurements and the odds of severe/very severe (GOLD grades 3 and 4) or mild/moderate stages (GOLD grades 1 and 2) of COPD across all tested models.

Severity Stages of COPD	Variables	[95% CI]	*p*-Value
Model 1			
Mild/moderate stages of COPD (GOLD grades 1 and 2) Severe/very severe stages of COPD (GOLD grades 3 and 4)	LV GLS TAPSE FAC-RV RV4CLS RVFWSL PASP S’RV	[−19.204–−18.506] [17.892–19.039] [29.105–32.994] [−16.032–−15.439] [−18.384–−17.821] [33.946–36.703] [11.173–12.203]	0.000 * 0.000 * 0.000 * 0.000 * 0.000 * 0.000 * 0.000 *
Model 2			
Mild/moderate stages of COPD (GOLD grades 1 and 2) Severe/very severe stages of COPD (GOLD grades 3 and 4)	LV GLS TAPSE FAC-RV RV4CLS RVFWSL PASP S’RV	[−19.204–−18.501] [17.887–19.051] [29.097–32.961] [−16.026–−15.462] [−18.380–−17.840] [33.922–36.698] [11.174–12.205]	0.000 * 0.000 * 0.000 * 0.000 * 0.000 * 0.001 * 0.000 *
Model 3			
Mild/moderate stages of COPD (GOLD grades 1 and 2) Severe/very severe stages of COPD (GOLD grades 3 and 4)	LV GLS TAPSE FAC-RV PASP S’ RV	Y (severity stages of COPD) = 28.901 + 1.542 × GLS Y (severity stages of COPD) = 16.245–0.903 × TAPSE Y (severity stages of COPD) = 8.176–0.263 × FAC-RV Y (severity stages of COPD) = −7.855 + 0.214 × PASP Y (severity stages of COPD) = 9.241–0.823 × S’ RV	0.000 * 0.000 * 0.000 * 0.001 * 0.000 *

Model 1: univariate model. Model 2: multivariate model including the biomarker and age, sex, and active smoking as covariates. Model 3: binomial logistic regression model, CI = 95% confidence interval; LV GLS—global longitudinal strain of the LV, RV—right ventricle, S’ RV—systolic longitudinal contraction of the RV, TAPSE—systolic motion of the tricuspid annulus plane, FAC—fractional aria change, RVFWSL—RV free wall longitudinal strain, RV4CSL—RV four chamber longitudinal strain, PASP—pulmonary artery systolic pressure; * significant result: *p* < 0.05.

## Data Availability

The raw data involved in this study can be obtained upon reasonable request addressed to Alexandra-Lucia Pop (2203alexandra@gmail.com).

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
