# Peer review of "Right Ventricular Strain and Left Ventricular Strain Using Speckle Tracking Echocardiography—Independent Prognostic Associations in COPD Alongside NT-proBNP"

_diseases, 2025, doi:10.3390/diseases13100344_

Round 1

Reviewer 1 Report

Comments and Suggestions for Authors

COPD is an important medical and social problem. 
Comments:
1. The information content of table 1 can be improved. There is insufficient clinical data on patients with COPD. Data on spirometry, concomitant diseases, frequency of exacerbations, and therapy should be added.
2. How was the healthy group recruited? Did they have any medical conditions and were they taking any therapy?
3. It is not clear why there were such small groups? COPD is not a rare disease, but this disease is heterogeneous, so the number of patients should be larger so that this heterogeneity can be taken into account, including the severity of the disease, the presence and frequency of exacerbations. For example, there are only 4 patients in the GOLD1 group. How can these data be analyzed?
4. When evaluating echocardiography, the contribution of concomitant cardiovascular diseases is not clear
5. In Tables 3-5, it is not clear which variables have statistically significant differences (there are several variables, for which values is there a statistically significant difference?).
6. The novelty and practical significance of the research is unclear. What does this study add to the existing body of knowledge? How can the data practically be used? Wouldn't it be easier to determine the severity of COPD based on spirometry data? If severity means something more than the stage of COPD, then it would be logical to add multidimensional scales such as BODE, eBODE, CODEX and others to the analysis.

Author Response

1. The information content of table 1 can be improved. There is insufficient clinical data on patients with COPD. Data on spirometry, concomitant diseases, frequency of exacerbations, and therapy should be added.

Response 1: Thank you for bringing this to our attention. We agree with this comment. Therefore, we have made the change mark in red in the revised manuscript. This change can be found on page 7, Table 2.

2.  How was the healthy group recruited? Did they have any medical conditions and were they taking any therapy?

Response 2: Thank you for bringing this to our attention. The healthy group was recruited from patient registries and physician referrals, through collaboration with general practitioners (GPs). Individuals who met the study criteria were identified and included after obtaining informed consent. It consisted of perfectly healthy patients with no medical condition, with no respiratory or cardiovascular symptoms, without any comorbidities or treatment.

3. It is not clear why there were such small groups? COPD is not a rare disease, but this disease is heterogeneous, so the number of patients should be larger so that this heterogeneity can be taken into account, including the severity of the disease, the presence and frequency of exacerbations. For example, there are only 4 patients in the GOLD1 group. How can these data be analyzed?

Response 3: Thank you for pointing this out. We agree with this comment. The number of patients included in the study is low considering the disease's global prevalence. However, at the medical center where this study was conducted, the incidence of this condition was not very high. Unfortunately, this is the number of patients who consented to participate in the study over the course of one year. Currently in Romania, COPD is not diagnosed well enough, with most cases being diagnosed at GOLD grade 2 or later stages. GOLD grade 1 is most often diagnosed accidentally, as most patients neglect their respiratory symptoms. We agree that statistical analysis cannot be performed on a sample of four people, but patients in GOLD stage 1 did not represent our analysis group. For complex analyses, we combined GOLD stages 1 and 2, and GOLD stages 3 and 4, to obtain statistically significant parameters.   4. When evaluating echocardiography, the contribution of concomitant cardiovascular diseases is not clear.

Response 4: Thank you for pointing this out. With regard to cardiovascular complications, there were no notable changes in the patients included in the study from an ultrasound perspective. Patients with severe systolic dysfunction (EF < 40%) were excluded from the study, and patients with ischemic heart disease (IHD) presented minor kinetic disorders, with EF > 50% in 90% of the patients included. For this reason, no statistically significant differences were recorded between the two groups in terms of LVEF. Patients with arrhythmias did not show significant changes in LA or RA volumes and function compared to those in the control group.

5. In Tables 3-5, it is not clear which variables have statistically significant differences (there are several variables, for which values is there a statistically significant difference?).

Response 5: Thank you for pointing this out. We agree with this comment. Therefore, we have made the change. We added Table 3 and Table 5 (pages 8 and 9), highlighting the statistical analysis performed between the COPD group and the control group. In another table (Table 2, Table 4, and Table 6, pages 8, 9, 10), we added only COPD patients, according to the severity of the disease. In those Tables, the “P” values actually represent the combination of COPD GOLD1 + 2 and 3 + 4, respectively.

6. The novelty and practical significance of the research is unclear. What does this study add to the existing body of knowledge? How can the data practically be used? Wouldn't it be easier to determine the severity of COPD based on spirometry data? If severity means something more than the stage of COPD, then it would be logical to add multidimensional scales such as BODE, eBODE, CODEX and others to the analysis.

Response 6: Spirometry is a crucial test for diagnosing and assessing the severity of COPD. However, in many cases, spirometry alone is not enough. Over time, some concerns have been raised about the reliability of spirometry. Schermer et al. reported a small but statistically significant difference between lung function values obtained by spirometry performed in a doctor's office and those obtained in a pulmonary function laboratory [1]. There remains a persistent problem with the inconsistent use and interpretation of spirometry results by physicians, even though scientific societies maintain that COPD diagnosis and treatment should only be initiated after objective spirometry testing and interpretation by specialized clinicians. A significant proportion of COPD diagnoses in primary care are false positives, with the use of spirometry to confirm the diagnosis being limited [2]. Echocardiographic evaluation often reveals cardiovascular complications specific to advanced stages of the disease, such as systolic dysfunction of the RV and LV, PH, and sometimes even chronic pulmonary heart disease. Because of this fact, we believe this article is practically important, as it highlights a warning sign for pulmonologists and cardiologists. To prevent the development of irreversible cardiovascular complications associated with COPD, advanced echocardiographic assessments and NT-proBNP testing are necessary. A value above 325 pg/mL strongly suggests an already advanced stage of the disease. To use the multidimensional indices analysis (BODE scales and other scales), we would have had to follow the patients in several stages (upon inclusion in the study and during the follow-up period). However, this study is an observational one.

  1. Schermer, T.R. Validity of Spirometric Testing in a General Practice Population of Patients with Chronic Obstructive Pulmonary Disease (COPD). Thorax 2003, 58, 861–866, doi:10.1136/thorax.58.10.861.
  2. Vila, M.; Sisó-Almirall, A.; Ocaña, A.; Agustí, A.; Faner, R.; Borras-Santos, A.; González-de Paz, L. Prevalence, Diagnostic Accuracy, and Healthcare Utilization Patterns in Patients with COPD in Primary Healthcare: A Population-Based Study. npj Prim. Care Respir. Med. 2025, 35, 17, doi:10.1038/s41533-025-00419-9.

Reviewer 2 Report

Comments and Suggestions for Authors
  1. This study has investigated primarily cardiac dysfunction-related indices in a relatively small group of COPD patients recruited from both respiratory and cardiac hospital services. The patients were divided into four sub-groups by severity, and indices compared between groups, as well as with a control group. Although not immediately relevant to the title, quite a number of inflammatory and pro inflammatory blood cells and proteins were also measured and related predominantly to COPD severity. Oxidant/antioxidant balance was also measured in the blood and related essentially to both parts of this matrix.
  1. I found the paper interesting, but very long and overcomplicated and not easy to digest. It needs a lot of simplifying. I would suggest focusing only on the prime outcome of cardiac measurements against COPD severity, and perhaps the oxidant story but again limited to its relationship with heart function. The rest is rather superfluous and not really novel; we know that infective inflammation increases with severity of COPD and especially in those with frequent exacerbations.
  1. The Abstract needs to be shortened and simplified, and I would suggest dropping all the echocardiographic acronyms and instead summarizing a broad-brush summary of how cardiac dysfunction relates to COPD severity, with perhaps a particular mention of pre-BNP levels as most people are already very aware of this in heart failure.
  1. The authors` overview of the pathophysiology of COPD is far from ideal or sufficiently accurate, and needs to be rewritten. To summarise: this is an airway  disease with luminal innate inflammatory activation to be sure, much of it secondary to bacterial colonization and subsequent overt infection. But the fundamental process from an early stage, is airway remodeling involving epithelial basal cell gene reprogramming (oxidants may be a major instigator here), growth factor production, EMT and sub epithelial myofibroblast proliferation, huge overproduction of modified matrix proteins and the thickening of all the tissue layers. This in turn results in airway luminal narrowing and ultimately destruction, especially in the smaller airways although it is a pan-airway pathology. Emphysema in this condition occurs late, and is secondary to bronchiolar air trapping, and indeed the emphysema is limited to those peri-bronchial alveolar areas. This needs to be distinguished from pan-acinar emphysema much is rare, and more related to a destructive imbalance in inflammatory cell control.
  1. The potential role of VEGF in COPD is grossly overdone, and I would suggest dropping it.
  1. When the rather niche echocardiographic indices are introduced, the focus should be much more on what they mean rather than technical details.
  1. There should be a strong possibility of co-linearity between quite a lot of the measures being assessed against COPD. I would suggest that it would help the reader for some work to be done to highlight the most informative and more independent one or two indices and relegate a lot of the noise and detail to a supplement. As far as I can see, a measurement of pre-BNP as an index of pulmonary hypertension and secondary right heart strain might be sufficient This would be an important, coherent and clinically useful message from the paper. At the moment the most important messages are rather overwhelmed by either irrelevant or tautologous information.
  1. I thought the oxidant part was quite neat, but one doesn't need a blood test to tell how severe COPD is. Symptoms, history and lung function tests are sufficient. On that issue, the paper is rather bereft of lung function data and each of the subgroups should have their spirometric measurements given and it would also be of value to give their diffusing capacities and lung volumes in the demographic and physiological table(s).
  1. I was struck by the relatively low level of smoking history even in the more severe group. Do you think the reports here, from patients themselves I presume, are accurate? Ideally, there should be a smoking but physiologically-normal group, especially for interpretation of the oxidant data.
  1. I have to question the discussion on hypoxia in this population studied. This is rare in COPD until extremely severe. Could there be confounding by the co-morbidity of obstructive sleep apnea which would be relatively common in such a population?
  1. No mention is given of the frequent difficulty of getting good cardiac echo-quality in the COPD population, especially if there is co-morbid emphysema. What was the actual experience like in this patient population?
  1. If as I am suggesting the paper becomes more focused, then much of its excessive length could/should be pruned.

Author Response

1. This study has investigated primarily cardiac dysfunction-related indices in a relatively small group of COPD patients recruited from both respiratory and cardiac hospital services. The patients were divided into four sub-groups by severity, and indices compared between groups, as well as with a control group. Although not immediately relevant to the title, quite a number of inflammatory and pro inflammatory blood cells and proteins were also measured and related predominantly to COPD severity. Oxidant/antioxidant balance was also measured in the blood and related essentially to both parts of this matrix.

Response 1: Thank you for your comment. We have also modified the title of the article, as it was not particularly relevant to the content.

2. I found the paper interesting, but very long and overcomplicated and not easy to digest. It needs a lot of simplifying. I would suggest focusing only on the prime outcome of cardiac measurements against COPD severity, and perhaps the oxidant story but again limited to its relationship with heart function. The rest is rather superfluous and not really novel; we know that infective inflammation increases with severity of COPD and especially in those with frequent exacerbations.

Response 2: Thank you for your comment. We simplified the original version of the article, hoping it will be easier to read and understand.

3. The Abstract needs to be shortened and simplified, and I would suggest dropping all the echocardiographic acronyms and instead summarizing a broad-brush summary of how cardiac dysfunction relates to COPD severity, with perhaps a particular mention of pre-BNP levels as most people are already very aware of this in heart failure.

Response 3: Thank you for pointing this out. We agree with this comment. Therefore, we have made the change regarding the abstract length and content.

4. The authors` overview of the pathophysiology of COPD is far from ideal or sufficiently accurate, and needs to be rewritten. To summarise: this is an airway  disease with luminal innate inflammatory activation to be sure, much of it secondary to bacterial colonization and subsequent overt infection. But the fundamental process from an early stage, is airway remodeling involving epithelial basal cell gene reprogramming (oxidants may be a major instigator here), growth factor production, EMT and sub epithelial myofibroblast proliferation, huge overproduction of modified matrix proteins and the thickening of all the tissue layers. This in turn results in airway luminal narrowing and ultimately destruction, especially in the smaller airways although it is a pan-airway pathology. Emphysema in this condition occurs late, and is secondary to bronchiolar air trapping, and indeed the emphysema is limited to those peri-bronchial alveolar areas. This needs to be distinguished from pan-acinar emphysema much is rare, and more related to a destructive imbalance in inflammatory cell control.

Response 4: Thank you for pointing this out. We agree with this comment. Therefore, we have made the change regarding the pathophysiology of COPD as you suggested, marked in red (on page 2 and 3).

5. The potential role of VEGF in COPD is grossly overdone, and I would suggest dropping it.

Response 5: Thank you for your comment. We dropped the VEGF parameter from the article.

6. When the rather niche echocardiographic indices are introduced, the focus should be much more on what they mean rather than technical details.

Response 6: Thank you for your comment and suggestion. I have modified this aspect in the text, marked in red on page 5.

7. There should be a strong possibility of co-linearity between quite a lot of the measures being assessed against COPD. I would suggest that it would help the reader for some work to be done to highlight the most informative and more independent one or two indices and relegate a lot of the noise and detail to a supplement. As far as I can see, a measurement of pre-BNP as an index of pulmonary hypertension and secondary right heart strain might be sufficient This would be an important, coherent and clinically useful message from the paper. At the moment the most important messages are rather overwhelmed by either irrelevant or tautologous information.

Response 7: Thank you for your suggestions. I simplified the old version and added most of the data in a Supplementary Material. And, as you suggested, we focused carefully on cardiac dysfunction related to COPD severity and on the independent prognostic role of NT-proBNP as an indicator of pulmonary hypertension and secondary right heart strain.

8. I thought the oxidant part was quite neat, but one doesn't need a blood test to tell how severe COPD is. Symptoms, history and lung function tests are sufficient. On that issue, the paper is rather bereft of lung function data and each of the subgroups should have their spirometric measurements given and it would also be of value to give their diffusing capacities and lung volumes in the demographic and physiological table(s).

Response 8: Thank you for your comment. I totally agree. The part of the article related to oxidative stress is now secondary. With regard to lung function, in Table 7 we have added the spirometric measurements performed for each subgroup.

9. I was struck by the relatively low level of smoking history even in the more severe group. Do you think the reports here, from patients themselves I presume, are accurate? Ideally, there should be a smoking but physiologically-normal group, especially for interpretation of the oxidant data.

Response 9: Thank you for your comment. I totally agree. The Information regarding the number of packs of cigarettes smoked was obtained from patients through medical history, and there is a possibility of minimizing this aspect, given that patients in severe stages of the disease are no longer active smokers. Regarding your suggestions, another smoking but physiologically normal group would have been useful in assessing oxidative stress in smokers/non-smokers. For this reason, the part of the article related to oxidative stress is now secondary, given that the information and statistical analysis are incomplete.

10. I have to question the discussion on hypoxia in this population studied. This is rare in COPD until extremely severe. Could there be confounding by the co-morbidity of obstructive sleep apnea which would be relatively common in such a population?

Response 10: Thank you for your comment. I completely agree. Hypoxia is a result of advanced stages of the disease. Overlap Syndrome (COPD and OSA) is a common condition in pulmonary disorders. However, from the time patients were included in the study, those presenting more than three of the specific symptoms of Obstructive Sleep Apnea, such as intermittent hypoxia, fragmented sleep, snoring, daytime sleepiness, morning headaches, or concentration problems, were excluded.   11. No mention is given of the frequent difficulty of getting good cardiac echo-quality in the COPD population, especially if there is co-morbid emphysema. What was the actual experience like in this patient population?

Response 11: Thank you for your comment. I completely agree. Obtaining good echocardiography can be difficult in COPD patients due to factors like air trapping and lung hyperinflation, which create poor acoustic windows for ultrasound waves. The challenge increases with the severity of the disease. For this reason, we use contrast echocardiography in some patients, especially those with severe COPD, to overcome these limitations. Specific adjustments to patient positioning and prolonged image acquisition timing were used to enhance image quality.

12. If as I am suggesting the paper becomes more focused, then much of its excessive length could/should be pruned.

 Response 12: Thank you for your comment. We updated the paper length, and the most important information is summarized in this article. We added Supplementary Material to reduce the complexity of the article.

Reviewer 3 Report

Comments and Suggestions for Authors

one-unit increase in Caspase-3 corresponds to a 0.002 (p=0.009) increase in the number of exacerbations: Should present as NNH or for 1 increase in AECOPD

Abstract: Results part presented in messy way. Seems presenting different outcomes in chaotic order. Not sure if they want to study COPD or CVS association.

Any sample size calculation?

A multiple linear regression model demonstrated that with a one-unit increase 302
in CRP, the number of exacerbations increases by 0.011 (p=0.007): Should present as NNH or for 1 increase in AECOPD

Table 5 very busy, suggest trim it and present main findings as main table and the rest in supplementary table

Need to be cauthious for the ROC which demonstarted very little sample size with the curve not smooth

Discussion and conclusion rather lenghty, can shorten

Author Response

Comment 1: one-unit increase in Caspase-3 corresponds to a 0.002 (p=0.009) increase in the number of exacerbations: Should present as NNH or for 1 increase in AECOPD.

Response 1: Thank you for pointing this out. We agree with this comment. Therefore, we have made the change. A multiple linear regression model demonstrated that for every one increase in AECOPD, Caspase-3 increases by 50.2 pg/mL (p = 0.009).

Comment 2: Abstract: Results part presented in messy way. Seems presenting different outcomes in chaotic order. Not sure if they want to study COPD or CVS association.

Response 2: Thank you for pointing this out. We agree with this comment. Therefore, we have reworded the content of the abstract, focusing on the COPD study.

Comment 3: Any sample size calculation?

Response 3:

 sample size calculation

Comment 4: A multiple linear regression model demonstrated that with a one-unit increase 302
in CRP, the number of exacerbations increases by 0.011 (p=0.007): Should present as NNH or for 1 increase in AECOPD

Response 4: Thank you for pointing this out. We agree with this comment. After applying a multiple linear regression model, we found that an increase in AECOPD by one unit was not statistically significant for CRP, and therefore, this information was removed from the article.

Comment 5: Table 5 very busy, suggest trim it and present main findings as main table and the rest in supplementary table.

Response 5: Thank you for pointing this out. We agree with this comment. Therefore, we have made the change. As you suggested, the table has been reorganized, leaving only the elements that are significant to this study (Table 8, Page 12). The extended version can be found in the Supplementary Material (Table S3).

Comment 6: Need to be cauthious for the ROC which demonstarted very little sample size with the curve not smooth.

Response 6: Thank you for pointing this out. We agree with this comment. The appearance of the ROC curve is because there are different numbers of patients included in each GOLD stage. If we had had an equal number of patients in each category, the ROC graph would have been much more uniform.

Comment 7: Discussion and conclusion rather lenghty, can shorten.

Response 7: Thank you for pointing this out. We agree with this comment. Therefore, we have made the change. We have shortened those two chapters, Discussions and Conclusions, as can be seen on pages 17 and 21, respectively.

(x) The English could be improved to more clearly express the research.

Response: The article will be sent to the authorized service for English and table corrections after the reviewers approve the updated version of the text.

Round 2

Reviewer 1 Report

Comments and Suggestions for Authors

Although the authors answered my questions, not all of the answers are in the text of the article. It is recommended to add to the text of the article the information that the authors provided in response to my questions, for example, how the statistical difference in severity in the groups of patients with COPD was calculated. Why was it necessary to divide into all 4 stages, rather than make 1-2 and 3-4 stages if the sample is so small? How to interpret the data on NT-proBNP levels for advanced stages if, in real practice, the difference between stages 3 and 4 of COPD is large enough, but in the current study the sample was small to assess the differences?

Author Response

Although the authors answered my questions, not all of the answers are in the text of the article. It is recommended to add to the text of the article the information that the authors provided in response to my questions, for example, how the statistical difference in severity in the groups of patients with COPD was calculated. Why was it necessary to divide into all 4 stages, rather than make 1-2 and 3-4 stages if the sample is so small? How to interpret the data on NT-proBNP levels for advanced stages if, in real practice, the difference between stages 3 and 4 of COPD is large enough, but in the current study the sample was small to assess the differences?

Response:  Thank you for pointing this out. We agree with this comment. Therefore, we have made the change. We added to this article the information provided in the first set of questions.

            In the following sections, we will attempt to address the questions that have been raised.

            The statistical difference in severity among COPD patient groups was analyzed using Kendall's correlation coefficient because these groups involve ordinal variables and scalar, numerical variables. We initially chose to retain the GOLD classification recommended by specialist guidelines. However, based on the statistically significant results obtained, we then continued with advanced statistical analysis (univariate and multivariate analysis), combining GOLD groups 1 and 2, and GOLD groups 3 and 4. This approach has been described in the literature and other recent clinical studies, with a reduced number of patients [1].

  1. Abdelalim, M.A.F.; Khalil, M.A.; Sharshr, R.S.; Abdelzaher, A.H. The Role of Transthoracic Echocardiography in Evaluating Right Ventricular Parameters in Chronic Obstructive Pulmonary Disease. Egypt J Bronchol 2024, 18, 71, doi:10.1186/s43168-024-00325-7.

            According to the literature, the difference in NT-proBNP levels between COPD stages 3 and 4 is generally not significant enough to serve as a reliable marker for distinguishing between them. While NT-proBNP levels do increase with overall COPD severity (comparing moderate to severe/very severe), this trend is broad rather than precise for differentiating stage 3 from stage 4, which is a known limitation [2–4]. The information provided in previous COPD guidelines is described below.

NT-proBNP level by GOLD stages ABCD based on 2011 and 2017 guidelines

GOLD stages

A

B

C

D

2011 guidelines

n

286

384

56

325

NT-proBNP level

575.4 ± 652.7

664.3 ± 1032.8

634.5 ± 692.7

568.6 ± 931.3

2017 guidelines

n

310

533

32

176

NT-proBNP level

580.5 ± 652.0

574.9 ± 728.6

629.1 ± 731.0

758.4 ± 1519.9

However, no significant differences in plasma NT-proBNP levels were observed between patients with stage III and stage IV disease (P = 0.310) [4].

  1. Abdelalim, M.A.F.; Khalil, M.A.; Sharshr, R.S.; Abdelzaher, A.H. The Role of Transthoracic Echocardiography in Evaluating Right Ventricular Parameters in Chronic Obstructive Pulmonary Disease. Egypt J Bronchol 2024, 18, 71, doi:10.1186/s43168-024-00325-7.
  2. Su, X.; Lei, T.; Yu, H.; Zhang, L.; Feng, Z.; Shuai, T.; Guo, H.; Liu, J. NT-proBNP in Different Patient Groups of COPD: A Systematic Review and Meta-Analysis. COPD 2023, Volume 18, 811–825, doi:10.2147/COPD.S396663.
  3. Chi, S.Y.; Kim, E.Y.; Ban, H.J.; Oh, I.J.; Kwon, Y.S.; Kim, K.S.; Kim, Y.I.; Kim, Y.C.; Lim, S.C. Plasma N-Terminal Pro-Brain Natriuretic Peptide: A Prognostic Marker in Patients with Chronic Obstructive Pulmonary Disease. Lung 2012, 190, 271–276, doi:10.1007/s00408-011-9363-7.
  4. Labaki, W.W.; Xia, M.; Murray, S.; Curtis, J.L.; Barr, R.G.; Bhatt, S.P.; Bleecker, E.R.; Hansel, N.N.; Cooper, C.B.; Dransfield, M.T.; et al. NT-proBNP in Stable COPD and Future Exacerbation Risk: Analysis of the SPIROMICS Cohort. Respir Med 2018, 140, 87–93, doi:10.1016/j.rmed.2018.06.005..

Reviewer 2 Report

Comments and Suggestions for Authors
  1. In this fairly rapid turnover response, the title is improved and a new section added to the Introduction but this really makes it only longer and more complex and more difficult to read. They should cut out all the stuff about the complexity of cellular information which isn't terribly relevant although potentially interesting in the broader sense, and stick to big picture concepts such as inflammation on the one hand and remodeling on the other. Inflammation is within the airway lumen essentially, at least until there is gross infection. In the earlier stages of COPD and pre-COPD, the airway wall is relatively acellular allowing for the expansion of its volume.
  2. As previously suggested there needs to be a complete rewrite and simplification of the whole paper. I would suggest a more thorough reading and integration of the previous comments that were made, and a real attempt at converting these into something more meaningful and readable.
  3. The authors keep using the term “predictors” when what they mean are “associations” of their biochemical and cardiac functional measures with lung function.
  4. The most useful content and the most interesting is the cardiac associations as summed up in their introductory main purpose of the study. Why not essentially stick to this element?
  5. On line 67, the authors make the comment that there is, quite rightly, an increase in bacterial load in COPD which increases with severity. They imply that this is broadly based but that is not the case; the predominant organism is non typeable Haemophilus influenzae. I would not go overboard on this, but they might want to consider recent information on why this is so, and this relationship to increased expression of PAF receptors by the epithelium.
  6. On line 127, that there is a rather unusual expression of “lungs fill up”. I have no idea what that means, and in general I would keep away from emphysema in this paper as you do such a late expression of COPD pathology, and just stick to the complexities of airway disease which are quite sufficient.

Author Response

1. In this fairly rapid turnover response, the title is improved and a new section added to the Introduction but this really makes it only longer and more complex and more difficult to read. They should cut out all the stuff about the complexity of cellular information which isn't terribly relevant although potentially interesting in the broader sense, and stick to big picture concepts such as inflammation on the one hand and remodeling on the other. Inflammation is within the airway lumen essentially, at least until there is gross infection. In the earlier stages of COPD and pre-COPD, the airway wall is relatively acellular allowing for the expansion of its volume.

Response: Thank you for pointing this out. We agree with this comment. Therefore, we have made the change from the introduction. And we focused, as you suggested, on inflammation and airway remodeling.

2. As previously suggested there needs to be a complete rewrite and simplification of the whole paper. I would suggest a more thorough reading and integration of the previous comments that were made, and a real attempt at converting these into something more meaningful and readable.

Response: Thank you for your suggestions. We agree with this comment. We have revised the article once again, simplifying its complexity and reducing its length. As you suggested, we have omitted much of the information and focused our attention on ultrasound measurements, NT-proBNP, and just a few details about oxidative stress in relation to cardiac function. To further simplify the results chapter, we have added information related to clinical, demographic, and anthropometric data, as well as participants' medications, in the Supplementary Materials. However, we have also tried to retain the content and add the requirements provided by the other reviewers.

3. The authors keep using the term “predictors” when what they mean are “associations” of their biochemical and cardiac functional measures with lung function.

Response: Thank you for pointing this out. We agree with this comment. Therefore, we have made the change in the text.

4. The most useful content and the most interesting is the cardiac associations as summed up in their introductory main purpose of the study. Why not essentially stick to this element?

Response: Thank you for your suggestions. We agree with this comment. We have revised the article once again, and we hope that it is the correct version.

5. On line 67, the authors make the comment that there is, quite rightly, an increase in bacterial load in COPD which increases with severity. They imply that this is broadly based but that is not the case; the predominant organism is non typeable Haemophilus influenzae. I would not go overboard on this, but they might want to consider recent information on why this is so, and this relationship to increased expression of PAF receptors by the epithelium.

Response: Thank you for pointing this out. Therefore, we have made the change in the introduction, marked in red.

6. On line 127, that there is a rather unusual expression of “lungs fill up”. I have no idea what that means, and in general I would keep away from emphysema in this paper as you do such a late expression of COPD pathology, and just stick to the complexities of airway disease which are quite sufficient.

Response: Thank you for your suggestions. We have revised the article once again and have decided not to include information about emphysema to reduce the complexity of the work.

Reviewer 3 Report

Comments and Suggestions for Authors
  1. Please include sample size calculation somewhere such as appendix if not yet done
  2. If ROC not looking good, please  consider to supplementary information

Author Response

1. Please include sample size calculation somewhere such as appendix if not yet done.

Response: Thank you for pointing this out. We agree with this comment. Therefore, we have made the change and added the sample size calculation to the Supplementary Materials.

2. If ROC not looking good, please  consider to supplementary information.

Response: To simplify the work, we decided to focus on the two ROC curves that highlight the predictive values of biological parameters and ultrasound measurements for advanced stages of the disease. The appearance of the ROC curve is due to the varying numbers of patients included in each GOLD stage. If we had had an equal number of patients in each category, the ROC graph would have been much more uniform.

Below we have added additional information related to ROC curves.

The ROC curve is created by plotting true positive rates (or sensitivity) against false positive rates (or 1-specificity) using different cut-off levels for a continuous predictor to forecast a binary outcome. Each point on the curve shows a balance, or trade-off, between sensitivity and specificity for a specific diagnostic or prognostic threshold of the continuous predictor. The area under the receiver operating characteristic (AUC) curve is commonly used in medicine to summarize how well a continuous diagnostic or predictive marker can identify or predict a binary or dichotomized outcome, such as having or not having a disease [5.6].

      NT-proBNP levels predict severe/very severe stages of COPD with a sensitivity (Se) of 52% and a specificity (Sp) of 88%. NT-proBNP is a test with high specificity, being very useful for diagnosing advanced COPD, as specificity is inversely proportional to the false positive rate.

      CRP levels predict severe/very severe stages of COPD with a sensitivity of 95.7% and a specificity of 71.9%. CRP has high sensitivity and can be considered a screening test, increasing the likelihood of correctly identifying patients with severe/very severe COPD who have an inflammatory syndrome. It provides a positive result in patients with COPD .

      RV4CLS, RVFWSL, and PASP exhibited high sensitivity but low specificity. These three echocardiographic parameters demonstrate high sensitivity and can serve as screening tests, increasing the likelihood of accurately identifying patients with severe or very severe COPD who have subclinical right ventricular dysfunction and pulmonary hypertension. They yield positive results in patients with COPD.

  1. Ho, K.M. Effect of Non-Linearity of a Predictor on the Shape and Magnitude of Its Receiver-Operating-Characteristic Curve in Predicting a Binary Outcome. Sci Rep 2017, 7, 10155, doi:10.1038/s41598-017-10408-9.
  2. Ho, K.M. Effect of non-linearity of a predictor on the shape and magnitude of its receiver-operating-characteristic curve in predicting a binary outcome. Sci Rep7, 10155 (2017). https://doi.org/10.1038/s41598-017-10408-9

Round 3

Reviewer 1 Report

Comments and Suggestions for Authors

The authors have made corrections to the text of the article, which has improved its quality.

Author Response

The authors have made corrections to the text of the article, which has improved its quality.

Response: Thank you for accepting our article in this form for publication.

Reviewer 2 Report

Comments and Suggestions for Authors
  1. The authors have improved this paper substantially by focusing on the indices of cardio-dysfunction in persons with COPD compared to normal in different levels of severity. However, the manuscript still needs substantial improvement and especially in simplifying the analysis and focusing on the key findings.
  2. What is important here is the difference between normal and the mildest degree of COPD which is rather ignored, and only then the differences between severity levels. What is also ignored is any attempt to personalize this in the sense of giving some sort of variance within groups, IE what is the least and greatest level of dysfunction. In the grade one COPD, what percentage have abnormal data and how many still within normal range?
  3. I cannot see these tests being used in any screening way for COPD diagnosis or severity assessment; that is based on symptomatology, exacerbation rates and lung function. What is important is the recognition that there may also be pulmonary hypertension even at a relatively early stage with secondary right heart strain. Whether it is then worth looking for that or not is a secondary issue, unless it is going to end to management in any way which at the moment is doubtful.
  4. There is no point in analyzing what the cut off at 50% of severity is. There is no point in analyzing all of COPD versus normal. All analysis should include normal control values. There is no point in trying to use these data as predictions as it they are cross-sectional. In a longitudinal setting it might well be worth assessing whether right heart strain early on is a predictor of survival but the authors are not in a position to do that.
  5. It should be more emphasized that the best index of cardiac or PH issues seems to be NT-BNP, which is just a blood test routinely available. That should be a major conclusion.
  6. There are some important findings and conclusions in this paper but the presentations need to be much simpler and the message clearer and not contrived to be more impressive than they actually are.  Simplicity and clarity is everything.
  7. L238, some superfluous words, and I suggest starting from the word ”Only”; L421, I am not sure what this means, “demographics” presumably; L424, the first line of this paragraph is just not necessary.

Author Response

  1. The authors have improved this paper substantially by focusing on the indices of cardio-dysfunction in persons with COPD compared to normal in different levels of severity. However, the manuscript still needs substantial improvement and especially in simplifying the analysis and focusing on the key findings.

Response: Thank you for all your comments. We appreciate your interest in this work. The chapter dedicated to statistical analysis has been revised and adapted to meet your requirements as well as those of the other two reviewers. They agreed with this final aspect of the statistical analysis. If we were to remove certain tables at this point, it would contradict the fact that the other two reviewers approved the final version of the thesis as suitable for publication.

2. What is important here is the difference between normal and the mildest degree of COPD which is rather ignored, and only then the differences between severity levels. What is also ignored is any attempt to personalize this in the sense of giving some sort of variance within groups, IE what is the least and greatest level of dysfunction. In the grade one COPD, what percentage have abnormal data and how many still within normal range?

Response: Since only four patients are included in GOLD grade 1, we cannot answer this question. The small sample size results in a lack of statistical significance, which is why this information was not included in the article. Currently in Romania, COPD is often underdiagnosed, with most cases diagnosed at GOLD grade 2 or later stages. GOLD grade 1 is most frequently diagnosed by accident, as many patients ignore their respiratory symptoms.

3. I cannot see these tests being used in any screening way for COPD diagnosis or severity assessment; that is based on symptomatology, exacerbation rates and lung function. What is important is the recognition that there may also be pulmonary hypertension even at a relatively early stage with secondary right heart strain. Whether it is then worth looking for that or not is a secondary issue, unless it is going to end to management in any way which at the moment is doubtful.

Response: Based on the data presented in the article, due to the small number of patients in GOLD 1, it is not possible to determine whether subclinical ventricular dysfunction or HTP is present in the early stages of the disease (GOLD 1). In GOLD 1 grade, we have only four patients included. In GOLD grades 1 and 2, only 5 out of 32 patients have PAPS above 37.5, and 7 patients have NTproBNP levels above 325 pg/mL (the maximum value is 570 pg/mL). The results are not statistically significant due to the small sample size.

4. There is no point in analyzing what the cut off at 50% of severity is. There is no point in analyzing all of COPD versus normal. All analysis should include normal control values. There is no point in trying to use these data as predictions as it they are cross-sectional. In a longitudinal setting it might well be worth assessing whether right heart strain early on is a predictor of survival but the authors are not in a position to do that.

 Response: You are right with these statements, as it is a cross-sectional study; no predictions can be made about survival. As you suggested, we added the normal control values to the tables. We kept the tables analyzing the two study groups—the control group and the COPD group—because, from our perspective, they also contain essential information: only after achieving statistical significance between the two groups did we proceed with the multivariate analysis of the previously identified significant parameters.

5. It should be more emphasized that the best index of cardiac or PH issues seems to be NT-BNP, which is just a blood test routinely available. That should be a major conclusion.

 Response: Thank you for your comment. We have modified the final conclusion, focusing on NP-proBNP.

6. There are some important findings and conclusions in this paper but the presentations need to be much simpler and the message clearer and not contrived to be more impressive than they actually are.  Simplicity and clarity is everything.

Response: Thank you for this observation. We have tried to reduce the complexity of the thesis, as per your request. However, we would like to mention that the other two reviewers had other suggestions related to the article (additions associated with AECOPD and statistical analysis) that we would like to keep. Considering that they have accepted the current form of the article.

7. L238, some superfluous words, and I suggest starting from the word ”Only”; L421, I am not sure what this means, “demographics” presumably; L424, the first line of this paragraph is just not necessary.

 Response: Thank you for your feedback; we have implemented the changes you requested.

Round 4

Reviewer 2 Report

Comments and Suggestions for Authors

Thanks you for the work you have done to improve this paper so much.